# β-Aminobutyric Acid Effectively Postpones Senescence of Strawberry Fruit by Regulating Metabolism of NO, H₂S, Ascorbic Acid, and ABA

Lei Wang [1,*], Jingru Liu [1], Meilin Li [2], Li Liu [1], Yonghua Zheng [3] and Hua Zhang [1]

[1] College of Agriculture and Agricultural Engineering, Liaocheng University, Liaocheng 252000, China; lilylcu112@163.com (J.L.); liuli13505407902@163.com (L.L.); zhanglcu@163.com (H.Z.)

[2] College of Food Science, Shenyang Agricultural University, Shenyang 110866, China; liml@syau.edu.cn

[3] College of Food Science and Technology, Nanjing Agricultural University, Nanjing 210095, China; zyhnjau@163.com

[*] Correspondence: freshair928@163.com; Tel.: +86-635-8239763

**Abstract:** Current research is focused on the influence of β-aminobutyric acid (BABA) on the metabolism of nitric oxide (NO), hydrogen sulfide (H₂S), ascorbic acid, and abscisic acid (ABA) in strawberry fruit. The increases in ion leakage and malondialdehyde (MDA) concentration in strawberry fruit and the degradation of chlorophyll in the sepals of the fruit were markedly inhibited by BABA at 20 mM. BABA-immersed fruit exhibited lower activities and expressions of polygalacturonase (PG), pectinmethylesterase (PME), and ethylene biosynthetic enzymes compared to the control. Furthermore, BABA immersion evidently upgraded the metabolic levels of NO and H₂S, including the enzymatic activities and intermediary contents of metabolites, which collectively enhanced the levels of endogenous NO and H₂S contents in strawberry fruit. The high enzymatic activities and gene expressions of the AsA biosynthesis pathway jointly maintained AsA accumulation in the BABA-treated sample. The application of BABA led to a decrease in ABA concentration, which was associated with reduced activities and gene expression levels of key enzymes participating in ABA metabolism. Our experimental observations showed that immersion with BABA may be a highly promising means to delay senescence and reduce natural decay in strawberry fruit, and the alleviation in senescence using BABA may be attributed to the modulation of NO, H₂S, AsA, and ABA metabolism.

**Keywords:** BABA; strawberry; hydrogen sulfide metabolism; nitric oxide metabolism; ABA

## 1. Introduction

Strawberry (*Fragaria* × *ananassa* Duch.) is a highly favored berry fruit that is cherished for its delightful flavor profile, nutrient-dense composition, and potent antioxidant potential [1]. The fruit is harvested under high-temperature circumstances. Therefore, in most cases, the potent metabolism of harvested strawberry fruit accelerates its senescence process. Moreover, fresh strawberry fruit is liable to suffer from mechanical injury and pathogen infection owing to its delicate tissue, abundant nutrition, and high moisture level. The inevitable postharvest senescence of strawberry fruit and the consequent decline in disease resistance results in sizable reductions in fruit edibleness and continuous growth in economic losses [1]. However, the senescence process can be regulated by postharvest treatments in fruit. Postharvest treatments aimed at delaying the senescence process of fresh fruit have been receiving increasing attention from agricultural practitioners and researchers. With the increasing demand for high-quality agricultural products, practical attempts toward the mitigation of senescence are becoming more and more crucial and urgent. The sizable body of existing literature suggests that the senescence of strawberry fruit could be retarded by external treatments such as strigolactone, nitric oxide, and elevated CO₂ [2–4]. Natural senescence in plants and fruit is a systematic and elaborate

physiological and biochemical process, in which many metabolic pathways are involved. The purposes of postharvest preservation are to reduce decay, maintain edibleness, and enhance resistance in the fruit and vegetable industry. Meanwhile, there are also various forms of postharvest-preservation technologies; among them, immersion is a convenient postharvest-treatment method, which has considerable potential wide spread application within the fruit and vegetable industry. The application of multiple approaches illustrates the effectiveness of postharvest treatments in terms of maintaining postharvest quality, extending shelf life, and improving antibacterial activity in strawberry fruit. Regardless of the postharvest practices applied, the efficacy of alleviating senescence is the preference for horticultural products.

It is well documented that the metabolism of internal components in fruit is closely related to natural senescence during storage. Ethylene, NO, hydrogen sulfide, and abscisic acid (ABA) are ubiquitous endogenous-signaling molecules in plants [5]. The metabolic activities of these signaling molecules in plants have been extensively documented at virtually every stage of plant life cycles. What is particularly noteworthy is that the metabolisms of the aforementioned signaling molecules are crucial in enabling plants to withstand environmental stresses and to mitigate the harmful impacts these stresses impose. As plant hormones, ethylene and abscisic acid are deeply involved in the postharvest aging process of strawberry fruit. The reduction of ethylene in strawberry fruit can delay the softening process of the fruit and further extend its shelf life [6]. ABA serves multiple purposes in postharvest strawberries, including mediating stress responses and modulating fruit over-maturation and storage qualities [7]. Furthermore, its complex interplay with other phytohormones ensures a well-coordinated progression through the different stages of fruit development and maturation [8]. As a considerable antioxidant in plants, ascorbic acid (AsA) and its metabolism are deeply involved in the regulation of the fruit-ripening process and stress tolerance, particularly during the postharvest storage phase [9,10]. Meanwhile, the orchestrated regulation of signal molecules and functional metabolites within plants plays a critical role in the process of stress-resistance signaling transduction and gene-expression induction. Remarkable advancements have highlighted the close connection between adversity stress (including senescence) and the metabolism of signaling molecules and AsA [9,10]. Undertaking relevant research will significantly enhance our understanding of the regulatory mechanisms underlying fruit senescence and thereby provide a theoretical underpinning for enhancing plant tolerance. The investigation of Wang et al. confirmed that sodium hydrosulfide treatment delayed the senescence of mushrooms due to the regulation of ethylene metabolism [11]. Experimental research showed that the application of exogenetic elicitor enhanced the metabolism of NO in peach fruit, which corresponded to a preferable postharvest quality [12]. The review findings of Verslues and Zhu reveal a profound link between abscisic acid metabolism and the regulation of adversity stress in plants [13]. However, the above-mentioned studies did not address the systematic study of the exogenous elicitor on the metabolism of internal signaling molecules and ascorbic acid in postharvest fruit. The metabolic regulation of low-molecular-weight signaling molecules in fruit still lacks experimental evidence during storage. β-aminobutyric acid (BABA) is a high-efficiency natural plant metabolic compound with two functionalgroups. As an important ingredient of the signal-path-regulating defensive response, BABA is a priming activator that provides wide spectrum crop protection in many agricultural products [14,15]. Intensive studies revealed that BABA could enhance antioxidative capacity and resistant substance in cucumber fruit [16]. Further studies demonstrated that BABA treatment improved the physiological parameters of different plants [16,17]. Given the wide spread utilizations of BABA in agronomy, it is imperative to gain a comprehensive understanding of its functions across diverse plant species. Based on our prior findings, BABA has been found to effectively slow down postharvest senescence and boost antioxidant activity in sweet cherry fruit kept at 20 °C [18]. However, the impact of BABA treatment on the relationship between signaling molecules' metabolism and their role in postponing senescence in strawberry fruit during

postharvest storage is yet to be evaluated. Current research is focused on the influence of BABA on the metabolisms of ethylene, NO, hydrogen sulfide, ABA, and ascorbic acid in strawberry fruit during room-temperature storage.

## 2. Materialsand Methods

### 2.1. Fruit Pretreatment

Fruit (*F. ananassa* Duch. cv. Fengxiang) were hand-harvested gently from a farm in the suburb of Liaocheng at commercial maturity stage. Defective fruit were discarded. The fruit with uniform size and color were used for the present study on the day of harvest. All strawberries were partitioned into two batches of 120 fruits each, randomly. One batch was immersed in a solution containing 20 mM BABA for a duration of ten minutes on the basis of a screening experiment (0, 10, 20, 30, 50 mM), which certified that the specified concentration of BABA (20 mM) had optimal effectiveness without any adverse phytotoxic influences. The other batch submerged in sterile water acted as the control. Subsequently, all fruit were allowed to air-dry at ambient temperature (20 °C) for roughly two hours before being stored at 20 °C for a period of three days under a relative humidity of 90%. The sample of 20 fruits from each group was used for the measurements of ethylene production and rot index daily. Another sample of 10 fruits with rotten tissue removed from each treatment group was rapidly frozen in liquid nitrogen for daily measurements of physio-biochemical characteristics. The current experiment was conducted twice, and each treatment group of 10 fruits had three biological replicates at every sampling juncture.

### 2.2. Natural Decay Evaluation

The percentage of infected fruit was recorded daily from each group during the course of the experiment. The fruit-rot-severity grading scale was as follows: Level 0 signifies no visible signs of rot; Level 1 indicates a rotten area covering less than one quarter of the fruit's surface; Level 2 represents a decayed region spanning between one quarter and one half of the fruit's surface; and Level 3 denotes a rotten area surpassing more than half of the fruit's surface. The calculation of the rot index was based on the following formula: Rot index = $\sum$(rot level × quantity of fruit at corresponding rank)/(3 × total fruit number).

### 2.3. Quantificationsof MDA, Chlorophyll, Ion Leakage, and Ethylene

The specification of Wang et al. was adopted to quantify the MDA content and ion leakage of experimental fruit [16]. Fruit pulp (2.0 g) was homogenized in 15 mL TCA (trichloroacetic acid), and the mixture was subsequently centrifuged at 8000× $g$ (4 °C, 15 min). The absorbance of extractive supernatant was estimated spectrophotometrically at three different wavelengths (450, 532, and 600 nm). MDA content was calculated as $6.45 \times (Ab_{532} - Ab_{600}) - 0.56 \times Ab_{450}$. Three grams of cylindrical strawberry specimens (2 mm thickness) from the cuticular layer of ten fruits were immersed in deionized water (25 mL) for 10 min. The conductivity was recorded ($D_i$). After boiling for ten minutes, the mixture was cooled to 20 °C. The electroconductivity ($D_f$) was measured in deionized water (25 mL). The experimental result was obtained by following expression–ion leakage = $(D_i/D_f) \times 100\%$. Chlorophyll concentration of sepal was assayed based on this published article [18]. The sepal sample (0.5 g) was ground in the solution of ethanol. The result was obtained spectrophotometrically and recorded in $g \cdot kg^{-1}$.

Ethylene concentration of fruit sample was detected referring to the protocol of Zhu and Zhou [4]. Fruit sample (500 g) was incubated in a hermetic jar for two hours at room temperature. Ethylene concentration in the headspace gas was measured by withdrawing 1 mL samples from the jar and injecting them into Agilent 6890GC gas chromatograph. The chromatograph was equipped with GS-Q capillary column and flame ionization detector. The nitrogen ($N_2$) served as the carrier gas with a consistent flow rate of 40 milliliters per minute. Ethylene concentration was recorded as $\mu L \cdot kg^{-1}$ sample$\cdot h^{-1}$.

### 2.4. Quantifications of Chlorophyllase, PG, PME, ACS, and ACO Activities

The activity of chlorophyllase was quantified referring to the protocol of Li et al. [2]. The frozen specimen (0.5 g) was pulverized in phosphate buffer at pH 7.0 including PVPP and cysteine. The reaction system included PMSF, acetone, polyethylene glycol octylphenol ether, and 3 mL enzyme supernatant. The result was obtained spectrophotometrically at 667 nm. The quantity of enzyme that produced one nanomole of chloride was designated as one unit of chlorophyllase activity.

The quantifications of PG and PME activity were operated by our previously published article [18]. One µmol galacturonic acid generated per mg of protein per hour in the reaction system equates to a unit of PG enzyme activity. A unit of PME enzyme activity is equal to 1 mmol sodium hydroxide titrated $h^{-1}mg^{-1}$ protein in the reaction system.

The protocol of Zhu and Zhou was selected to assess the activities of ACS (1-aminocyclo propane-1-carboxylic acid (ACC) synthase) and ACO (ACC oxidase) [4]. Five grams of frozen fruit tissue were crushed in an ice-cold Hepes buffer solution to obtain the enzyme extract. The reaction system included dithiothreitol (0.25 mM), S-adenosyl-L-methionine (0.1 mM), pyridoxal phosphate (0.25 mM), and supernatant (2 mL). The ACS activity of the sample was recorded in microliter ethylene $kg^{-1}$ sample $h^{-1}$. Enzyme extract of ACO was derived by grinding five grams of frozen sample in ice-cold Tris buffer (pH7.5) solution. The reaction mixture consisted of glycerine, sodium ascorbate (15 mM), PVP (1%, $v/v$), and supernatant (2.5 mL). ACO activity of sample was recorded in microliter ethylene $kg^{-1}$ sample $h^{-1}$.

The method outlined by Bradford was employed to quantify the protein concentration in enzyme extracts derived from strawberry fruit [19].

### 2.5. Quantifications of $H_2S$ Metabolism-Related Enzyme Activities and Intermediary Metabolite Contents

The concentration of endogenous $H_2S$ was quantified according to the specification of Geng et al. [20]. The strawberry fruit pulp (2.0 g) was comminuted in 10 mL phosphate buffer (including 0.20 M AsA and 0.1 M EDTA), and then subjected to centrifugation at $12,000\times g$ for 15 min. Next, the supernatant was added to the mixture of zinc acetate (1 mM) and hydrogen chloride (1 mL, 1 M), and they were incubated for half an hour. The solution was used for spectrophotometric observation at 667 nm. The concentration of $H_2S$ was determined using a calibration curve of sodium sulfide. The extraction of cysteine in strawberry fruit was extracted by pulverizing pulp tissue (2.0 g) with 5 mL of ice-cold sodium phosphate buffer. The supernatant was collected by centrifuging homogenate at $9000\times g$ (4 °C). Cysteine content was evaluated according to the manufacturer's protocols of the detection Kit (Beijing Solarbio Science and Technology Co., Ltd., Beijing, China). The results were reported in terms of millimole per kilogram fruit sample. The activitiesof L-cysteine desulfhydrase (LCD) and D-cysteine desulfhydrase (DCD) were assessed by the direction of Geng et al. [20]. Strawberry fruit pulp (2.0 g) was comminuted in 20 mL of ice-cold 20 mM Tris-HCl. The extracts were centrifuged at $12,000\times g$ for twenty minutes at 4 °C. The reaction liquid consisted of 0.9 mL supernatant and 1.1 mL Tris-HCl (containing 0.8 mM L-cysteine and 2.5 mM DTT). The activities of LCD and DCD were evaluated spectrophotometrically at 670 nm by the release of hydrogen sulfide according to the calibration curve of sodium sulfide. One unit of LCD or DCD activity was defined as the amount of enzyme producing 1µmol of $H_2S$ production per min. The direction of Huang et al. was used to assess the activities of O-acetylserine thiolyase (OAS) and serine acetyltransferase (SAT) [3]. For OAS, the experimental strawberry sample (2.0 g) was comminuted in 8 mL 50 mM phosphate buffer (including 10 µM pyridoxal phosphate and 2 mM DTT). The extracts were centrifuged at $15,000\times g$ for 30 min at 4 °C. The reaction liquid consisted of 0.9 mL supernatant and 1.1 mL HEPES-KOH (containing 5 mM $Na_2S$, 10 mM O-acetyl-serine and 5 mM DTT). The reaction was terminated by adding 0.5 mL of 10% ($v/v$) TCA. One unit of OAS activity was specified as the quantity of enzyme bringing 1 µmol of cysteine production $min^{-1}$. For SAT, the experimental strawberry sample (2.0 g)

was comminuted in 8 mL 30 mM potassium phosphate buffer (including 10 μM L-serine, 5 mM sodium sulfide, 2 μM acetyl-CoA, and 1 μM cysteine synthase). The extracts were centrifuged at $15,000 \times g$ for thirty minutes at 4 °C. The reaction solution consisted of 1.8 mL supernatant, and 2.2 mL bovine serum albumin (10 mM). Reaction mixture was activated with acetyl-CoA and terminated with HCl. One unit of SAT activity was specified as the quantity of enzyme bringing 1 μmol of cysteine production $min^{-1}$. The results are expressed in U $g^{-1}$ of fruit sample.

### 2.6. Quantifications of NO Metabolism-Related Enzyme Activities and Intermediary Metabolite Contents

The experimental strawberry sample (4.0 g) was ground in 10 mL 30 mM acetic acid. The extracts were centrifuged at $10,000 \times g$ for twenty minutes at 4 °C. NO content was evaluated according to the manufacturer's protocols of the NO Kit (Beijing Solarbio Science and Technology Co., Ltd., Beijing, China). Experimental results were recorded as $μmol \cdot kg^{-1}$ fruit sample. The fruit pulp sample (4.0 g) was comminuted with 20 mL 40 mM HEPES-KOH (including 5 mM cysteine,1 mM DTT, 0.5 mM EDTA, and 2 mM PMSF). The extracts were centrifuged at $12,000 \times g$ for twenty minutes at 4 °C. The guidance of Huang et al. was adopted to assay nitrate reductase (NR) and nitric oxide synthase-like (NOSL) activities [3]. The fruit pulp sample (4.0 g) was pulverized in 20 mL 0.1 mM HEPES-KOH (including 2 mM DTT, 1 mM EDTA, 1 mM PMSF, 2 mM sodium molybdate, 2 mM leupeptin, and 10 mM flavin adenine dinucleotide). The extracts were centrifuged at $12,000 \times g$ for twenty minutes at 4 °C. One unit of NR activity was specified as the production of one nmol nitrite $s^{-1}$ under the conditions of the assay. One unit of NOSL activity was specified as thequantity of enzyme bringing 1 μmol of nitric oxide production $s^{-1}$. Results are documented in Units $g^{-1}$ of fruit pulp. The concentrations of arginine (Arg) and nitrite were quantitatively assessed by the direction of Huang et al. [3]. For Arg, the experimental sample (2.0 g) was comminuted in 10 mL of ice-cold 10% (*w/v*) trichloroacetic acid at 4 °C. The extracts were centrifuged at $12,000 \times g$ for twenty minutes at 4 °C. The supernatant was used for spectrophotometric observation at 530 nm. For nitrite, strawberry fruit pulp (2.0 g) was comminuted with a mixture of double-distilled water (2 mL) and saturated borax (0.4 mL) followed by a 15 min boiling-water bath. Next, the obtained solution was added to the mixture of zinc acetate (1 mL, 1 M) and potassium ferrocyanide (1 mL, 0.25 M), and they were centrifugated for twenty minutes at 4 °C. The supernatant was used for spectrophotometric observation at 450 nm. The calculated results are expressed as micromoles per kilogram of frozen pulp.

### 2.7. Quantifications of AsA Content, APX, MDR, and DDR Activity

AsA concentration in strawberry sample was judged based on the instruction of Liu et al. [21]. Five grams of frozen sample was ground with 15 mL oxalic acid. The result was obtained through titration with 2, 6-dichlorophenol indophenols and documented in g $kg^{-1}$ of pulp.

APX activity was measured using the outline of Liu et al. [22]. The reaction system included 50 μL enzyme suspension, 1.4 mL sodium phosphate buffer (50 mmol $L^{-1}$, pH 7.0), 50 μL AsA (9 mmol $L^{-1}$), and 10 μL hydrogen peroxide (30%, *v/v*). One unit of enzyme activity was designated as the quantity of enzyme that resulted in the decline at the absorbance of 290 nm spectrophotometrically per minute. The guidance of Liu et al. was used to measure monodehydroascorbate reductase (MDR) activity [21]. Two grams of frozen sample was mashed in 15 mL Tris-HCl buffer containing PVP (5%, *w/v*), DTT (1 mM), Triton X-100 (1%), and EDTA (5 mM) and then subjected to centrifugation at $15,000 \times g$. The reaction mixture consisted of ascorbate, HEPES-KOH buffer, ascorbate oxidase enzyme, NADH, and 0.6 mL enzyme supernatant. The result was obtained spectrophotometrically at 340 nm. The instruction of Liu et al. was used to estimate the dehydroascorbate reductase (DDR) activity [21]. Five grams of frozen sample was mashed in 15 mL 2-hydroxyethyl buffer. The reaction system included dehydroascorbate (10 mM), GSH (6 mM), ethylene

diamine tetraacetic acid (5 mM), HEPES-KOH (50 mM), and 0.5 mL enzyme collection. The above results are presented in U/g fruit pulp.

### 2.8. Quantifications of ABA Content, Abscisic Acid Aldehyde Oxidase (AAO), and 9-Cis-Epoxycarotenoid Dioxygenase (NCED) Activity

The accumulation of ABA in frozen strawberry sample was judged based on the criterion of Siebeneichler et al. [22]. Frozen tissue (1.5 g) was powdered completely in prechilled solution of methanol (80%). The extracts were centrifuged at $13,500 \times g$ for twenty minutes at 4 °C. The supernatant was filtered with 0.22 μM syringe filter (Biosharp, Hefei, China). After removing polar compounds through Sep-Pak C18 cartridge, the filtrate was detected according to the manufacturer's protocols of ABA enzyme-linked immunosorbent assay (Adanti Biotechnology Co., Ltd., Wuhan, China). Experimental result is documented in $\mu g \cdot kg^{-1}$ of fruit pulp. AAO activity was estimated by the direction of Carvajal et al. [23]. Two grams of frozen sample was mashed in 15 mL phosphoric acid buffer, and then subjected to centrifugation at $10,000 \times g$ for 15 min. According to the manufacturer's protocols of the AAO ELISA method, the result was obtained spectrophotometrically at 450 nm. One unit of AAO activity was specified as the production of one nmol abscisic acid under the conditions of the assay. NCED activity was evaluated according to the previous method [24]. Frozen sample (2.0 g) was mashed in 15 mL PBS buffer, and then subjected to centrifugation at $15,000 \times g$ for 20 min. According to the manufacturer's protocols of the ELISA kit (Mlbio, Shanghai, China), the result was obtained spectrophotometrically at 450 nm.

### 2.9. Quantifications of Gene Expression

RNA was extracted from strawberry sample following the procedure outlined by Wang et al. [16]. Fruit tissue (5.0 g) was crushed with liquid nitrogen, and the resultant powder was collected for RNA isolation. Total RNA was extracted from this powder using CTAB solution (Sigma, Tokyo, Japan) that contained 2% ($v/v$) β-mercaptoethanol. The extracted RNA was measured using a Thermo Nano Drop 2000 spectrophotometer (Thermo Fisher Scientific, Waltham, MA, USA). The synthesis of complementary DNAwas carried out by reverse transcription according to the manufacturer's guidelines (Yeasen Biotechnology Co., Ltd. Shanghai, China). Quantitative real-time PCR was conducted according to the guidelines of the GoTaq® qPCR Master Mix kit (Promega, Madison, WI, USA) using a biochemical reaction setup that comprised cDNA template, 2.5 μL of each corresponding primer, 5 μL double-distilled water, and 15 μL qPCR Master Mix. The primers' sequences in this present research were obtained according to the bioinformation of NCBI database and listed in detail in Supplementary Table S1. The comparative expression levels of the target gene and the internal reference gene were determined using $2^{-\Delta\Delta CT}$ method.

### 2.10. Statistical Analysis

The current study was designed and executed using a complete randomization approach. The obtained data were assessed statistically based on a two-factor ANOVA framework, with treatment and storage duration as independent variables. Differences between treatments were statistically compared using Tukey–Kramer multiple range tests at a significance level of $p < 0.05$.

## 3. Results

### 3.1. Influence of BABA on Natural Rot Index, Ion Leakageof Strawberry Pericarp, Concentrations of MDA and Chlorophyll, and Chlorophyllase Activity

The rot index in both the control and treated strawberry fruit displayed a gradual upward trend with the extension of storage duration (Figure 1A). The rot index of the control group was obviously higher than that of the treated group in the last two days of incubation. The results revealed that BABA immersion at 20 mM was extremely effective in reducing the natural rot index in strawberry fruit. The ion leakage and MDA concentration of postharvest strawberry fruit persistently increased during the entire incubation

(Figure 1A,B). Ion leakage and MDA concentration from the BABA-treated group were always lower than those in the control group. The chlorophyll concentration in the posthar-vest sample displayed a downward trend and had an abrupt decrease in untreated sepals (Figure 1C). As for the chlorophyllase activity of the BABA-treated sample, a continuous downtrend was found throughout the experiment (Figure 1D). Chlorophyllase activity in the control sample underwent one day of increase and a subsequent slight decrease. Sepals treated with BABA displayed a higher concentration of chlorophyll and a lower activity of chlorophyllase compared to the untreated control group.

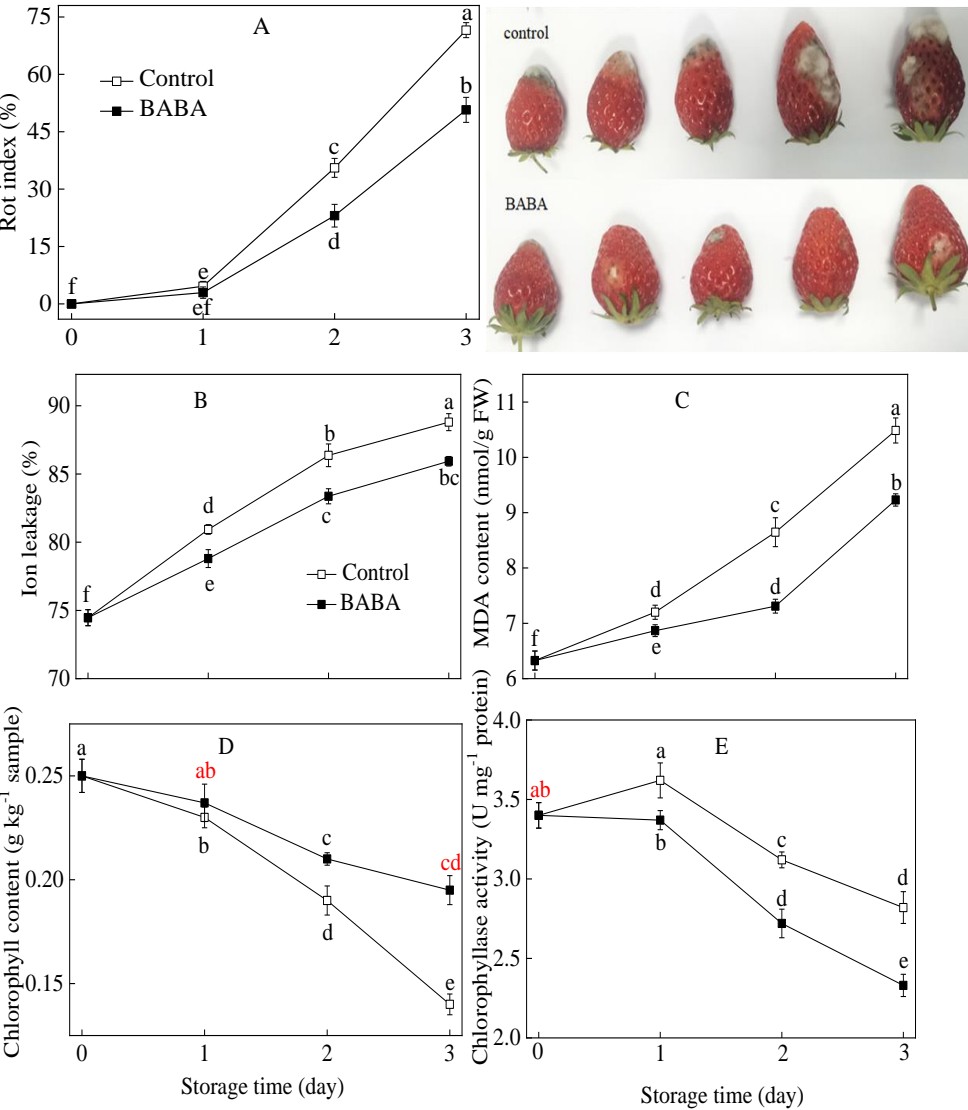

**Figure 1.** Rot index (**A**), ion leakage (**B**), MDA concentration (**C**), chlorophyllcontent (**D**), and chlorophyllase activity (**E**) in strawberry fruittreated with BABA at 20 mM for three days. Data are expressed as average values ± standard errors. In charts, different letters indicate the significant difference at $p < 0.05$.

### 3.2. Influence of BABA on Activities and Gene Expressions of PG, PME, ACS, and ACO, and Ethylene Concentration in Strawberry Fruit

The activity of PG in strawberry fruit displayed a gradual upward trend with the extension of storage duration (Figure 2A). The PME activity of postharvest strawberry fruit persistently increased during the entire incubation (Figure 2B). BABA prominently suppressed the rise of PG and PME activities starting from the first day of storage. The expression of *FaPG* was up-regulated at the second day of storage and down-regulated

afterwards in the control group. BABA repressed the expression of *FaPG* in strawberry fruit (Figure 2F). The expression level of *FaPME* in the control fruit initially increased significantly over the first 2 days before experiencing a slight decline. BABA decreased the transcript level of *FaPME* significantly on day 2 in comparison with the control (Figure 2F). As for the ethylene concentration of the fruit sample, a continuous growth trend was found throughout the experiment. BABA treatment effectively postponed the increase in ethylene concentration (Figure 2C). The activities of ACS and ACO in strawberry samples displayed similar gradual upward trends with the extension of storage duration (Figure 2D,E). BABA treatment effectively postponed the increase in the above-mentioned two biochemical parameters. BABA immersion resulted in lower expression levels of *FaACS* and *FaACO* in comparison with the control samples (Figure 2F).

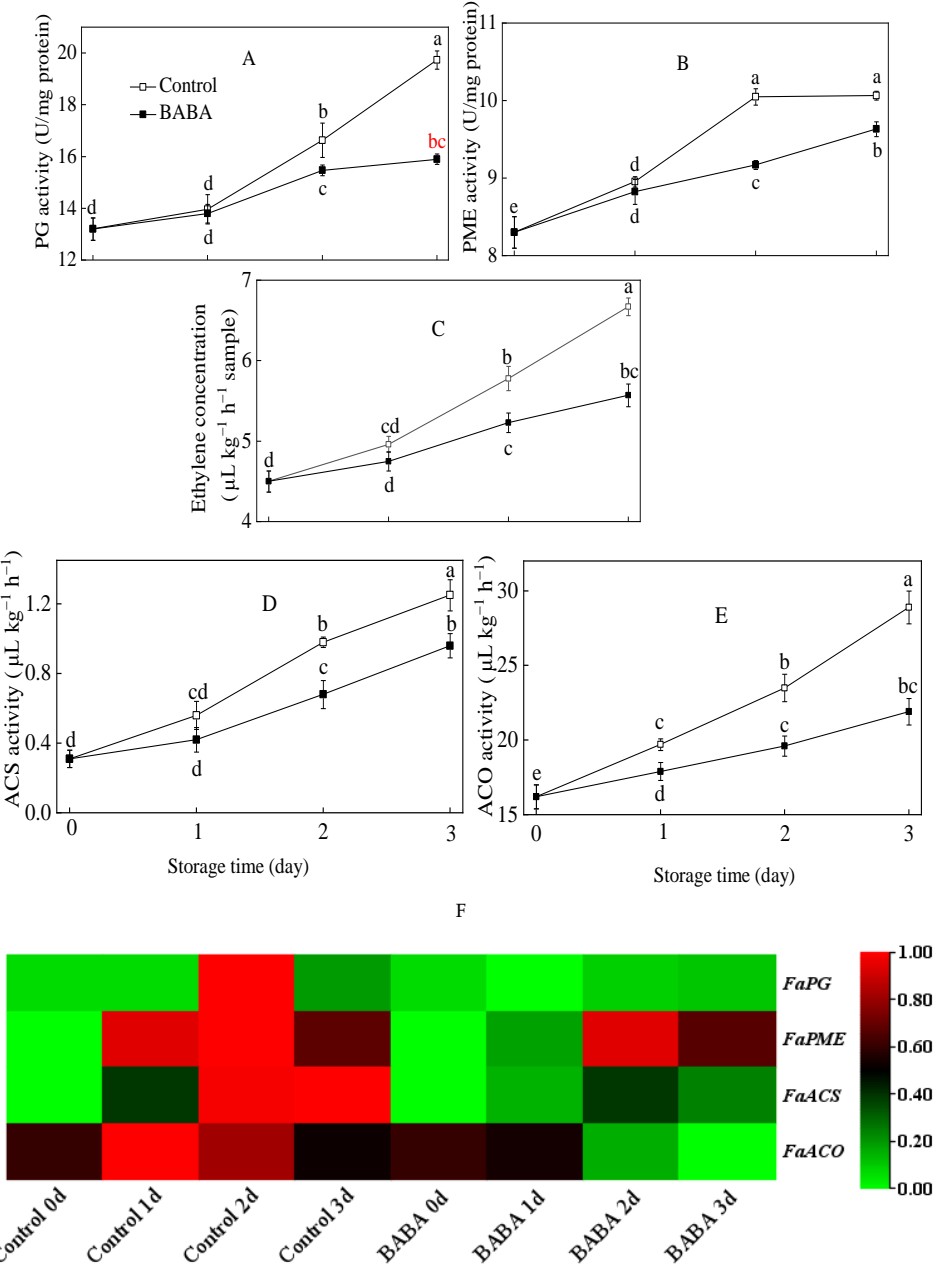

**Figure 2.** The activities of PG (**A**), PME (**B**), ACS (**D**), ACO (**E**), and ethylene concentration (**C**) and gene expressions of aforementioned enzymes (**F**) in strawberry fruit treated with BABA at 20 mM for three days. Data are expressed as average values $\pm$ standard errors. In charts, different letters indicate the significant difference at $p < 0.05$.

### 3.3. Influence of BABA on Parameters Associated with Hydrogen Sulfide Metabolism in Strawberry Fruit

The reduction in the content of Cys was observed in both the control and the treated strawberry fruit (Figure 3A). BABA application effectively retarded the reduction in Cys content. The DCD activity in the control strawberry fruit decreased slightly, whilst it reached its peak at 2 d and declined afterwards in the BABA-treated fruit (Figure 3C). The activities of LCD, OAS, and SAT in strawberry fruit progressively increased as the storage duration extended (Figure 3B,D,E). BABA application effectively accelerated the growth of LCD, OAS, and SAT activities. The $H_2S$ content increased continuously in all treatment groups during the incubation period (Figure 3F). BABA promoted the accumulation of $H_2S$ content in the strawberry fruit during postharvest storage.

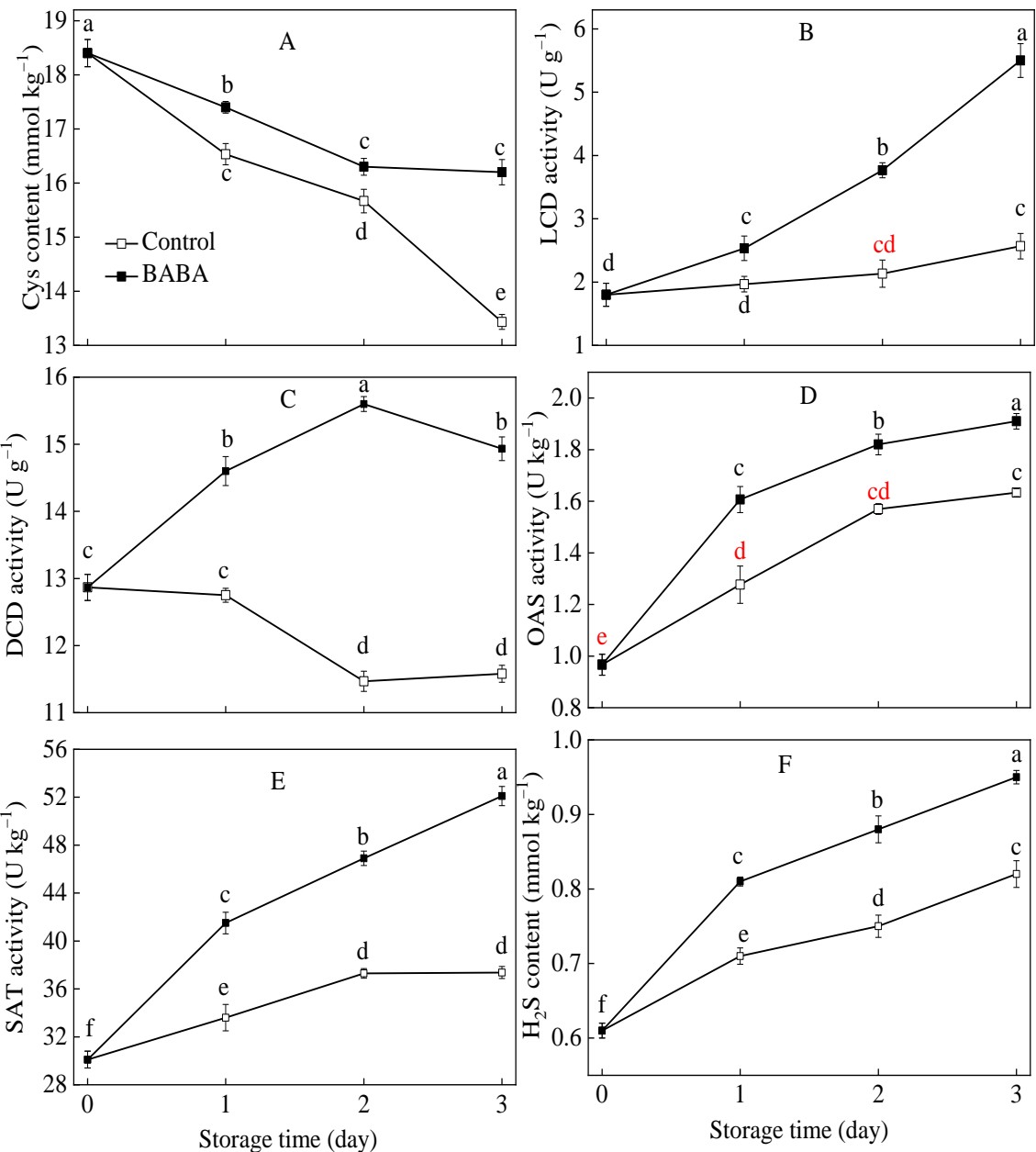

**Figure 3.** The contents of Cys (**A**), $H_2S$ (**F**), and activities of LCD (**B**), DCD (**C**), OAS (**D**), and SAT (**E**) in strawberry fruit treated with BABA at 20 mM for three days. Data are expressed as average values $\pm$ standard errors. In charts, different letters indicate the significant difference at $p < 0.05$.

### 3.4. Influence of BABA onParameters Associated with Nitric Oxide Metabolism in Strawberry Fruit

The enhancement of NR activity was monitored in strawberry fruit (Figure 4A). BABA promoted the increase in NR activity in treated strawberry fruit. The continuous increase in nitrite content was observed in the experimental sample. The control fruit displayed a gentle rise in nitrite content in comparison with the treated group (Figure 4B). The NOSL activity in the treated strawberry fruit increased continuously, whilst it reached its peak at 2 d and declined afterwards in the control treatment (Figure 4C). Arg and NO contents in the strawberry fruit increased along with storage duration. The contents of Arg and NO in the strawberry fruit displayed similar growing tendencies in all treatment groups (Figure 4D,E). BABA promoted the accumulation of Arg and NO contents in the strawberry fruit during postharvest storage.

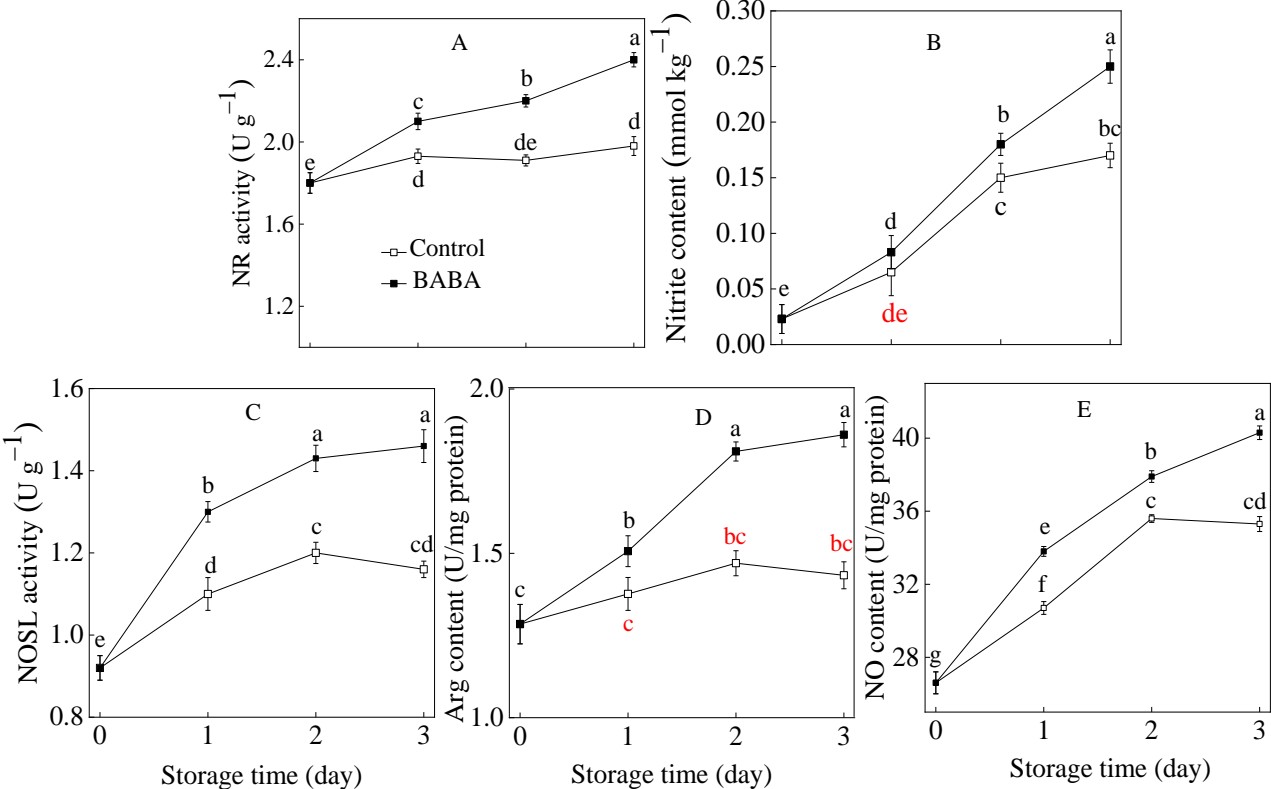

**Figure 4.** The activities of NR (**A**), NOSL (**C**), and contents of nitrite (**B**), Arg (**D**), and NO (**E**) in strawberry fruit treated with BABA at 20 mM for three days. Data are expressed as average values ± standard errors. In charts, different letters indicate the significant difference at $p < 0.05$.

### 3.5. Influence of BABA onParameters Associated with AsA Metabolism in Strawberry Fruit

AsA content and MDR activity showed declining trends during the storage period. The BABA treatment effectively postponed the decrease in the above-mentioned two parameters (Figure 5A,D). BABA immersion resulted in a slight increase in APX and DDR activities in the first day and experienced a gradual decline in the rest of the storage duration (Figure 5B,F). The BABA-treated strawberry fruit consistently exhibited higher APX and DDR activities compared to the control group. In both the control and the treated fruit, the expression levels of *FaAPX* and *FaMDR* gradually decreased over the course of three days of storage (Figure 5C,E). In the treated strawberries, *FaDDR* expression concentrations increased in the first two days of storage and then decreased slightly (Figure 5G). BABA enhanced the gene expression of three aforementioned enzymes in comparison with the control group during the course of the experiment.

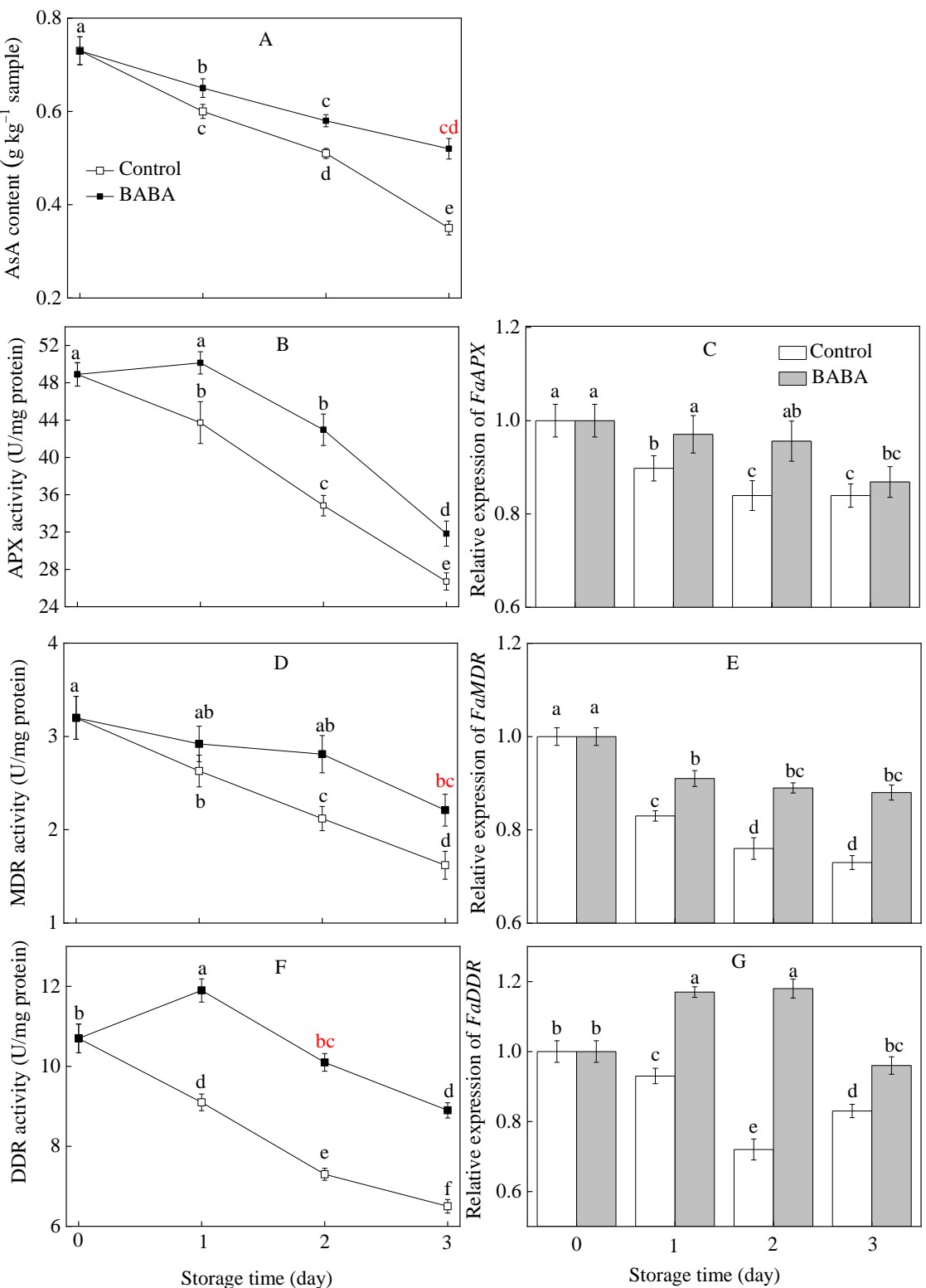

**Figure 5.** The content of AsA (**A**) and activities of APX (**B**), MDR (**D**), DDR (**F**), and gene expressions of *FaAPX* (**C**), *FaMDR* (**E**), *FaDDR* (**G**) in strawberry fruit treated with BABA at 20 mM for three days. Data are expressed as average values ± standard errors. In charts, different letters indicate the significant difference at *p* < 0.05.

### 3.6. Influence of BABA onParameters Associated with ABA Metabolism in Strawberry Fruit

The ABA concentration in the strawberry fruit showed an upward trend as the storage period was extended. The ABA content in the 20 mM BABA-treatment group was significantly lower than that in the control group (Figure 6A). The activities of NCED and AAO increased first and then declined afterwards. The use of BABA intervention effectively curbed the escalating tendencies of these two enzymes (Figure 6B,D). The expression profile of *FaNCED* in strawberries progressively escalated throughout the storage duration (Figure 6C). *FaAAO* expression in the control group increased dramatically and then decreased slightly (Figure 6E). Throughout the entire storage period, BABA application caused a significant down-regulation of *FaNCED* and *FaAAO* expression.

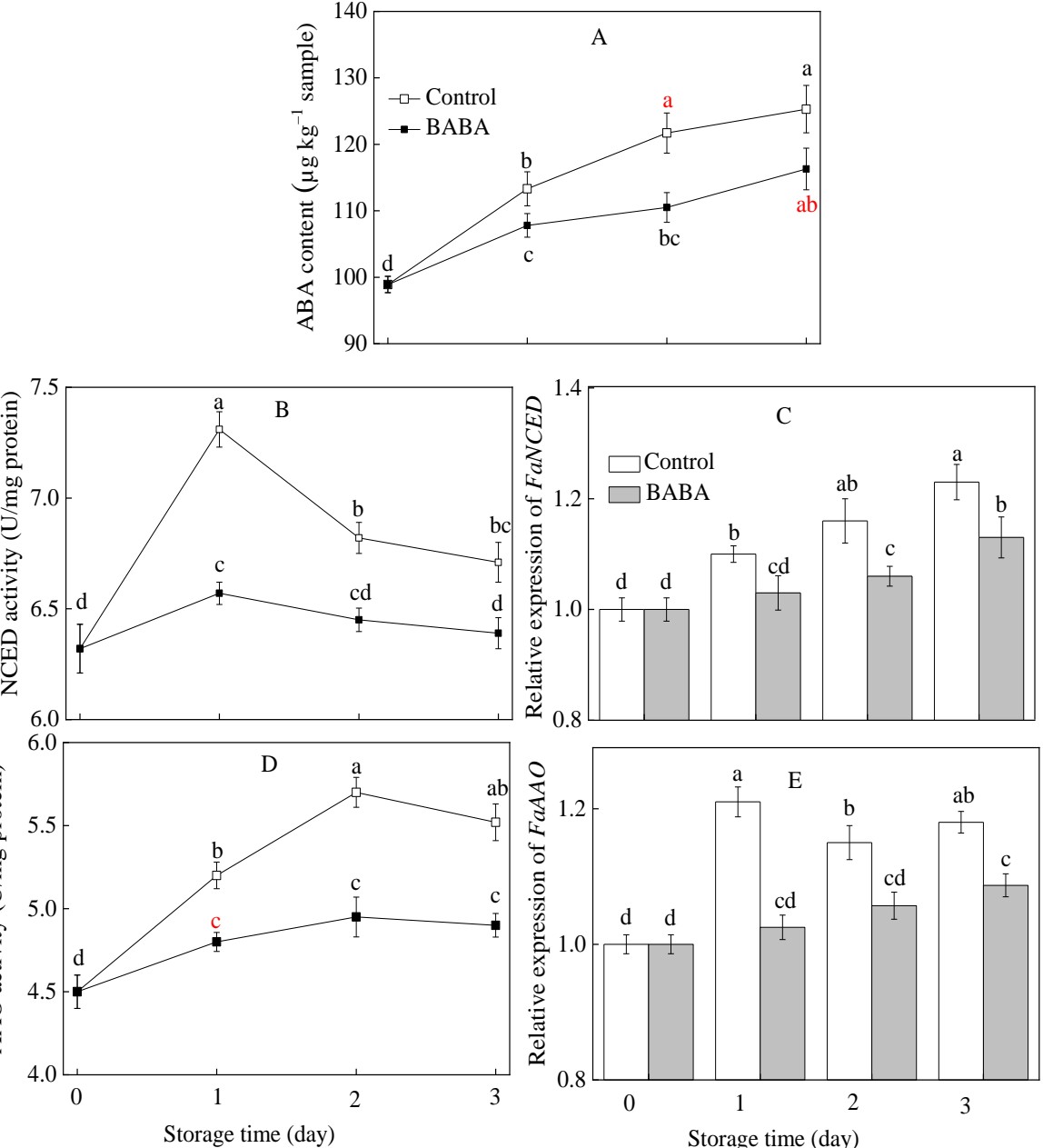

**Figure 6.** The content of ABA (**A**) and activities of NCED (**B**), AAO (**D**) and gene expression of *FaNCED* (**C**), *FaAAO* (**E**) in strawberry fruit treated with BABA at 20 mM for three days. Data are expressed as average values ± standard errors. In charts, different letters indicate the significant difference at *p* < 0.05.

## 4. Discussion

Fruit senescence, an inevitable biological process, directly affects the commercial value and disease resistance of fruits. Postharvest strategies that delay senescence have considerable potential for enhancing the fruit industry's productivity. Among these methods, BABA immersion has been employed across numerous plant species to serve various objectives, such as postponing senescence, reducing softening, and triggering defense responses [15,25]. The significance of endogenous signaling molecule metabolism in horticultural commodities is being increasingly recognized, as it plays a pivotal role in numerous physiological processes. Lipid peroxidation, free radical-induced oxidative stress, and fruit softening are the main factors involved in fruit senescence [26]. Senescent fruit is highly susceptible to microbiological infection, which causes economic losses and potential health problems in the fruit industry. In this study, it was observed that BABA (20 mM) treatment effectively retarded senescence and decreased natural decay in strawberry fruit.

Membrane damage causes the decline of membrane integrity, manifesting as elevated levels of conductivity and MDA contents [18]. MDA content, an indicator of cell oxidative damage, is correlated with fruit senescence [27]. Furthermore, the accumulation of MDA is detrimental to the cell membrane and the organelles of plant cells. Emerald green sepal is a characteristic trait indicating strawberry freshness. The browning of sepals in strawberry fruit during storage or transportation may lead to a decline in consumers' purchase intentions. Published evidence has elucidated that chlorophyllase is closely correlated with the decomposition of chlorophyll and plays a pivotal role in fruit senescence [28]. According to Hu et al., strawberry senescence is greatly implicated in MDA concentration [29]. The relevant results suggested that the application of hydrogen sulfide could down-regulate the accumulation of MDA production in strawberries. Yu et al. emphasized that the effective suppression of chlorophyllase in the leaves of tea could attenuate the degradation of chlorophyll [30]. In the present work, the ion leakage and MDA concentration of strawberry fruit increased with storage time, and the increase was effectively inhibited by BABA treatment. Meanwhile, the postharvest application of BABA substantially reduced the degradation of chlorophyll via the regulation of chlorophyllase in the sepal of strawberry fruit. This observation revealed that the delay of fruit senescence via BABA might be related to the reduced ion leakage, depressed chlorophyllase activity, and low level of MDA content.

As widely described in the existing literature, fruit softening is an irreversible physiological phenomenon in the process of fruit senescence [31,32]. The excessive softening in fruit increases the susceptibility of pathogen invasion and the possibility of postharvest decay [33]. It is widely accepted that a group of cell-wall-modifying enzymes plays a critical role in the process of fruit softening [31,34]. PG and PME play crucial coherent roles in cell-wall disassembly and consequent fruit softening [32]. PME acts as the catalyst of hydrolyzing cell-wall polygalacturonans. The subsequent products can be more easily degraded via PG. As an important hydrolase, PG has been found to promote the dissolution of pectin linked by $\alpha$-1, 4-glycosidic bonds [34]. Ethylene, as one of the most crucial signaling molecules in plants, participates in multitudinous biochemical processes including softening and senescence [35]. It is well documented that ACS and ACO are vital enzymes in the process of ethylene biosynthesis [36]. Recent evidence proposed that proper postharvest practice could alleviate fruit softening via the regulation of ethylene synthesis and cell-wall decomposition [37]. Our former studies observed that the process of softening was delayed effectively in sweet cherry fruit due to the inhibition of the activities of PG and PME [18]. The observation of Martínez and Civello confirmed that heat treatment reduced the process of strawberry-fruit softening due to the down-regulation of *FaPG* expression and the inhibition of PG activity [38]. In the present research, lower activities of PG, PME, ACS, and ACO were detected in BABA-immersed fruit. Moreover, BABA treatment down-regulated the expression of the above-mentioned genes in the whole incubation. Therefore, this detailed observation agreed with the records in the documents and found that BABA-treatmentdelayed strawberryfruit softening may be due to both down-regulated transcription levels and reduced enzyme activities [38].

Intracorporal nitric oxide in plants serves as a gasotransmitter involved in numerous physiological processes of plant life cycles, including maturation, senescence, and tolerance to adversity stresses [39]. Nitric oxide modulates the internal metabolic process of plants, and the metabolic process synchronously orchestrates the dynamic equilibrium of NO concentration. In plants, NO can be generated via non-enzymatic pathways such as the reduction of endogenous nitrite and the decomposition of nitrogen dioxide, as well as via enzymatic pathways such as the conversion of nitrite and arginine in the presence of critical enzymes [40]. NOSL catalyzes the formation of nitric oxide using arginine as a substrate. NR is involved in the conversion of nitrite to nitric oxide in plants. Scientific evidence revealed that NO has emerged as a characteristic signal molecule under the condition of adversity stresses in plants. Liu et al. verified that the increased activity of NR and consequent elevated NO levels relieved the development of senescence in pear fruit exposed to melatonin [40]. Research in strawberry fruit certified that the activity of NOSL and the content of arginine and nitric oxide were improved by the immersion of strigolactone, which ultimately maintained storage quality during storage [3]. In the current investigation, the application of BABA elevated the activities of NR and NOSL and maintained high levels of nitrite and arginine contents in strawberries during postharvest incubation. This investigation implied that the alleviation of senescence and natural decay following BABA immersion could be associated with the modulation of NO metabolism-related parameters in strawberry fruits.

Mounting compelling reports revealed that intracorporal hydrogen sulfide serves as a bioactive signal molecule involved in various developmental stages for plants, including postharvest senescence and biotic and abiotic stresses [20,41]. In plants, $H_2S$ is considered to be biosynthesized under the action of numerous enzymes, including LCD, DCD, OAS, and SAT. LCD and DCD catalyze the formation of hydrogen sulfide using L-/D-cysteine as substrates, respectively. OAS is involved in the conversion of inorganic sulfur to cysteine in plants. The resulting cysteine can be further converted to hydrogen sulfide under a cascade of enzymatic reactions [42]. SAT, a critical enzyme that regulates intracellular cysteine levels, promotes the generation of O-acetylserine using acetyl coenzyme A and L-serine as substrates. O-acetylserine promotes the decomposition of cysteine to produce hydrogen sulfide. Moreover, hydrogen sulfideis consumed by the reaction with O-acetylserine, which maintains the equilibrium state of hydrogen sulfide in the plant [3]. The changes in $H_2S$ levels are closely correlated with adversity stress (including postharvest senescence) in numerous horticultural plants. Li et al. observed that hydrogen sulfide treatment delayed the senescence of broccoli by modulating hydrogen sulfide metabolism-related enzymes (LCD and DCD) and consequent hydrogen sulfide concentration [43]. The discoveries obtained by Huang et al. verified that the high content of hydrogen sulfide attributed to high $H_2S$ metabolism-related enzymes (LCD, DCD, OAS, and SAT) activities and elevated level of Cys content played primary roles in maintaining the postharvest quality of strawberry fruits [3]. Our investigation certified that immersion with BABA observably promoted the activities of $H_2S$ metabolism-related enzymes and the content of intermediary metabolites (Cys), which jointly raised the content of endogenous hydrogen sulfide. Therefore, it is reasonable to infer that the mitigation of senescence and natural decay via BABA immersion could be attributed to the modulation of $H_2S$ metabolism-related parameters in strawberry fruit.

AsA, a powerful antioxidant and an indispensable nutrient, is fundamental to both plants and animals [44]. Ascorbic acid (AsA) not only serves as a regulator of oxidative stress responses, but it also plays an essential role in the synthesis of signaling molecules that significantly contribute to plant growth and defense mechanisms [45]. The concentration of AsA within fruit tissues serves a dual purpose, enhancing the fruit's nutritional attributes while concurrently reinforcing its capacity to withstand diverse environmental stress conditions [22]. Growing evidences point to the involvement of an array of recycling enzymes in the intricate metabolic pathways of AsA. The recycling process of AsA may play a more crucial role than the biosynthesis process in preserving AsA accumulation

during the postharvest stage of fruit. One pivotal enzyme, APX, performs a dual function: it not only detoxifies surplus hydrogen peroxide but also triggers the conversion of AsA to MDHA, which can subsequently be converted into dehydroascorbate via a process known as dismutation [21]. Concurrently, MDHA and dehydroascorbate can be re-converted to AsA through the enzymatic processes mediated via monodehydroascorbate reductase and dehydroascorbate reductase, respectively [44]. Numerous sources of credible proof have strongly indicated a close relationship between the senescenceprocess and AsA metabolic processes within fruit and vegetables. Luo et al. discovered that exogenous melatonin application led to an enhancement in the expression of genes associated with AsA metabolism, specifically *FaAPX*, *FaMDR*, and *FaDDR*, which consequently delayed the postharvest senescence of kiwifruit during storage [46]. Recent experimental findings disclosed that phenyllacticacid-dipping enhanced the postharvest quality of the winter jujube fruit, which was presumably attributed to the increased levels of enzymatic activities and gene expressions of *FaAPX*, *FaMDR*, and *FaDDR*, and succeeding enhancement in AsA accumulation [27]. In this study, the application of BABA led to heightened activities of the APX, MDR, and DDR in strawberry fruit during storage, concurrent with an up-regulation of their corresponding gene expressions. It is logical to deduce that the increased enzymatic activities and transcriptional level induced via BABA treatment contribute to higher AsA accumulations, which in turn help mitigate senescence processes and natural decay in strawberry fruit.

ABA, a potent signaling molecule, plays a pivotal role in the regulation of ripening and quality attributes in strawberry fruit. The finding that the increase in ABA content hastens postharvest fruit over-ripening confirms the proposal of its unique regulatory role in retarding strawberry fruit senescence [7]. Compelling research confirmed that NCED and AAO are involved in the process of ABA biosynthesis, which is essential for maintaining a critical balance within plants [47]. However, environmental stress and elicitor stimulation can affect ABA accumulation by regulating enzymes activities and gene expression. Xu et al. disclosed that the endogenous abscisic acid content and the expression levels of genes associated with ABA biosynthesis were decreased in response to UV-C in strawberry fruit during the later stage of storage [48]. Their experimental findings indicated that UV-C treatment potentially delays the senescence of strawberry fruit by modulating the metabolic pathways of ABA. In contrast, the findings of Carvajal et al. have ascertained that postharvest treatment could effectively reduce chilling injury in zucchini fruit by increasing the content of ABA [23]. This could be attributed to distinctions between fruit cultivars and tissue-specific variations. It was found in our study that BABA immersion decreased the concentration of ABA and decreased the activities and gene expression of NCED and AAO in strawberry fruit. Therefore, we speculate that the ABA metabolic changes triggered by BABA treatment play a role in delaying senescence in strawberry fruit when stored at room temperature.

## 5. Conclusions

In conclusion, the findings of this study indicated that BABA application delayed strawberry fruit senescence effectively duringstorage periods at 20 °C, potentially by depressing increases in ion leakage and MDA content, reducing the degradation of chlorophyll in fruit sepals, decreasing the concentrations of ABA and ethylene via the regulation of activities and corresponding gene expression, retarding fruit softening, and elevating the contents of NO, $H_2S$, and AsA; furthermore, BABA treatment inhibited postharvest natural decay in strawberry fruit. The current scientific observation implied that immersion with BABA may be a highly promising approach to delay senescence in strawberry fruit.

**Supplementary Materials:** The following supporting information can be downloaded at https://www.mdpi.com/article/10.3390/horticulturae10030218/s1, Table S1: Primer sequences for gene expression in qRT-PCR analysis [2,49,50].

**Author Contributions:** Conceptualization, supervision and writing-original draft preparation, L.W. and Y.Z.; methodology and investigation, J.L. and L.L.; writing-review and editing, M.L. and H.Z.; formal analysis, L.W.; project administration, H.Z. All authors have read and agreed to the published version of the manuscript.

**Funding:** This research was funded by Shandong Provincial Natural Science Foundation (ZR2021MC 185), National Natural Science Foundation of China (32001751), and Liaocheng University Student Innovation Training Program (CXCY2022297 and CXCY2022397). And The APC was funded by ZR2021MC185.

**Data Availability Statement:** Data are contained within the article and Supplementary Materials.

**Conflicts of Interest:** The authors declare no conflict of interest.

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
