# Peer review of "β-Aminobutyric Acid Effectively Postpones Senescence of Strawberry Fruit by Regulating Metabolism of NO, H2S, Ascorbic Acid, and ABA"

_horticulturae, doi:10.3390/horticulturae10030218_

Round 1

Reviewer 1 Report

Comments and Suggestions for Authors

The document addresses a topic that is already quite well known, given that strawberries have been the subject of much research to extend their useful life after harvest. The document shows an adequate wording to show the interaction of different physiological activities in the fruit and how they are affected by the treatment applied. But the document must be improved.

The consumer decides whether or not to buy a fruit based on its visual appearance, without the need to perform a biochemical test or ethylene production analysis, and also uses his or her sense of touch to evaluate the maturity or quality of the fruit through firmness, in the case of the strawberry, the texture says a lot about the quality of the fruit.

Because the monitoring of fruit deterioration was measured visually, the authors must incorporate photographs showing and comparing the treatment and the control group. This will allow a better understanding for the reader and provides certainty of the described effect. In addition, the photographs will allow us to show the color of the fruits, which can also be an indirect indicator of the physiology of the fruits or the presence of damage.

The authors did not evaluate the texture or firmness of the fruits instrumentally (using a texturometer or penetrometer), the authors must justify why this parameter was not evaluated given that the texture of the fruit is also an indirect indicator that consumers use to determine the freshness or shelf life of the fruit. In addition, texture is also related to the activity of enzymes related to the stability of pectin and the plant cell wall.

The authors must answer and justify with scientific support the following questions:

1.-Do the authors have to justify why the study only lasts 3 days and no more?

2.- Do strawberries only last 3 days of shelf life after harvest?

3.-Why is the study cut off after 3 days if the treatment is supposed to be aimed at extending the useful life of the strawberry?

4.- Is there no significant difference between the days of shelf life of treated and untreated fruits?

5.- Figure 1 A shows that the deterioration of the control fruit is greater than the fruit treated with BABA, however, the authors do not continue monitoring the treated fruit to demonstrate the effectiveness of the treatment. How do you justify the investment in treatment?

  6.- Why is there no photographic evidence of the fruits? Why was the texture of the fruits not measured?

7.- The authors must include comparative photographs of the fruits treated and not treated with BABA.

8.- Because ethylene production is used to demonstrate two physiological factors, the authors must broadly describe the methodology to quantify ethylene (detailed chromatographic conditions).

9.- The authors must amply justify why the monitoring of the useful life of treated and untreated fruits lasts the same time if the document aims to show the delay in senescence.

Author Response

The document addresses a topic that is already quite well known, given that strawberries have been the subject of much research to extend their useful life after harvest. The document shows an adequate wording to show the interaction of different physiological activities in the fruit and how they are affected by the treatment applied. But the document must be improved.

The consumer decides whether or not to buy a fruit based on its visual appearance, without the need to perform a biochemical test or ethylene production analysis, and also uses his or her sense of touch to evaluate the maturity or quality of the fruit through firmness, in the case of the strawberry, the texture says a lot about the quality of the fruit.

Because the monitoring of fruit deterioration was measured visually, the authors must incorporate photographs showing and comparing the treatment and the control group. This will allow a better understanding for the reader and provides certainty of the described effect. In addition, the photographs will allow us to show the color of the fruits, which can also be an indirect indicator of the physiology of the fruits or the presence of damage.

The authors did not evaluate the texture or firmness of the fruits instrumentally (using a texturometer or penetrometer), the authors must justify why this parameter was not evaluated given that the texture of the fruit is also an indirect indicator that consumers use to determine the freshness or shelf life of the fruit. In addition, texture is also related to the activity of enzymes related to the stability of pectin and the plant cell wall.

Response: Thank you very much for taking the time to review this manuscript. We have carefully considered the questions you raised and have responded to each of them point-by-point manner. The major revisions of the manuscript have been marked in red color. We provided comparative photographs in the revised manuscript. The main objective of the present study is to evaluate the impact of BABA treatment on the relationship between signaling molecules' metabolism and its role in postponing senescence in strawberry fruit during postharvest storage. As an important quality attribute, the texture of fruit is closely related to the fruit resistance to stress and the length of shelf life. Future studies may focus on the effects of BABA on the change of texture and pectin in strawberry fruit to achieve even greater benefits.

The authors must answer and justify with scientific support the following questions:

1.-Do the authors have to justify why the study only lasts 3 days and no more?

Response: Thanks for your constructive advice. Strawberry fruit has a soft texture and an extremely short shelf life. At room temperature storage conditions, strawberries are prone to spoilage during storage. Consumers typically avoid purchasing strawberries that are in a state of decay during the sales phase. In this experiment, a total of four days were covered from the time of harvest to the end of the experiment (0, 1, 2, 3 day). Considering the substantial decline in consumer appeal for strawberries that exhibit noticeable natural spoilage, which undermines both food safety and marketing value, our experimental process spanned a duration of four days.

2.- Do strawberries only last 3 days of shelf life after harvest?

Response: Thanks so much for your professional advice. Different strawberry cultivars, ranging from those with soft to firmer textures, exhibit varying shelf lives, and storage temperature significantly impacts the shelf life of strawberries. The storage period of strawberries at low temperature can even up to 15 days, which is also mentioned in many studies. In the present study, the strawberries used showed severe natural decay by the end of four days under room temperature storage conditions.

3.-Why is the study cut off after 3 days if the treatment is supposed to be aimed at extending the useful life of the strawberry?

Response: Thanks for your professional suggestions. Strawberry fruit has a short shelf life, especially under room temperature storage conditions. Different postharvest treatments can effectively enhance the stress resistance of strawberries through various regulatory mechanisms. Numerous studies have demonstrated that signaling molecules within plants (such as ethylene, NO, H2S, ABA, among others) are involved in the plant's regulatory processes in response to various adverse environmental conditions. The results of this study show that although BABA treatment does not completely inhibit postharvest natural decay in strawberries, it does increase the fruit's resistance to natural decay during storage by adjusting internal signaling molecules and concurrently delays the aging process. This finding provides a theoretical basis for the application of BABA in postharvest stage, which is the primary objective of this research. As the degree of decay increases, strawberries also lose their commercial value; thus, this study was carried out over duration of four days.

4.- Is there no significant difference between the days of shelf life of treated and untreated fruits?

Response: So appreciate your professional advice. The problem was similar to the previous problem. Post-harvest senescence in strawberry fruit is inevitable. Given the short shelf life and high susceptibility of strawberries to natural decay, this study's findings confirm that BABA treatment effectively delays the senescence of strawberry fruit and enhances their resistance against natural decay during post-harvest storage by modulating internal signaling molecules. The experiment has confirmed the effectiveness of BABA treatment in strawberry fruit; however, further research is required to elucidate the differences between the days of shelf life of treated and untreated fruits.

5.- Figure 1 A shows that the deterioration of the control fruit is greater than the fruit treated with BABA, however, the authors do not continue monitoring the treated fruit to demonstrate the effectiveness of the treatment. How do you justify the investment in treatment?

Response: Thanks for your professional advice. Natural decay in strawberry fruit occurs concurrently with senescence under room temperature storage conditions, which significantly impairs the economic value of the fruit. Mounting compelling evidences indicate that BABA is identified as a potent signaling molecule that involved in enhancing plant resistance to diverse biotic and abiotic stresses. Our experimental observations manifested that the alleviation of senescence and natural decay by BABA may be attributed to the modulation of NO, H2S, AsA, and ABA metabolism in strawberry fruit. Although BABA treatment does not completely suppress post-harvest natural decay, it does enhance stress resistance. This finding aligns with results from other researches, collectively substantiating the effectiveness of BABA in postharvest applications and providing a theoretical foundation for its use in postharvest field.

6.- Why is there no photographic evidence of the fruits? Why was the texture of the fruits not measured?  

Response: So appreciate your professional advice. We provided comparative photographs in the revised manuscript. The current research focused on the influence of BABA on the metabolisms of ethylene, NO, hydrogen sulfide, ABA, and ascorbic acid in strawberry fruit during room temperature storage. The results indicate that BABA treatment enhances enzyme activities and gene expression in the metabolic processes involving these signaling molecules. The texture of fruit is a critical quality attribute, and its changes during storage are closely related to the fruit resistance to stress and the length of shelf life. The study on the change of texture in strawberry fruit is also very important. Future studies may focus on the effects of BABA on the change of texture in strawberry fruit to achieve even greater benefits.

7.- The authors must include comparative photographs of the fruits treated and not treated with BABA.

Response: Thanks for your professional advice. We accepted the reviewer's comments and included comparative photographs in the revised manuscript.

8.- Because ethylene production is used to demonstrate two physiological factors, the authors must broadly describe the methodology to quantify ethylene (detailed chromatographic conditions).

Response: Thanks for your professional suggestions. We accepted the reviewer's comments and supplemented detailed chromatographic conditions in the revised manuscript. Such modification made the statements of present study more accurate.

9.- The authors must amply justify why the monitoring of the useful life of treated and untreated fruits lasts the same time if the document aims to show the delay in senescence.

Response: So appreciate your professional comments. Postharvest senescence of strawberry fruit involves various aspects, such as increased decay, fruit softening, and quality deterioration. This study demonstrated the effectiveness of BABA treatment by showing that it reduces natural decay, softening, and oxidative damage in strawberry fruit during storage. Meanwhile, internal levels of nitric oxide, hydrogen sulfide, ascorbic acid, and abscisic acid within strawberry fruit also play a role in regulating the aging process. Without monitoring for the same length of time, it would be difficult to pinpoint exactly when the benefits of the treatment begin to manifest or to quantify the extent of the delay in senescence. If the treated strawberries do show a delay in senescence, this should become evident when they maintain freshness longer than their untreated counterparts during the same period of monitoring.

Besides, a few type errors and font have been corrected in revised version.

Reviewer 2 Report

Comments and Suggestions for Authors

In this manuscript, the authors showed that BABA delayed the senescence of strawberry fruits by regulating metabolism of NO, H2S, ascorbic acid, and ABA. These results contribute to our understanding postharvest physiology of strawberry fruits. However, there are some issues in this manuscript as described below.

1. In Introduction, it is necessary to describe the current knowledge regarding the involvement of ethylene and ABA in the aging of strawberry fruits.

2. Duncanʼs multiple range test is applied for statistical analysis. However, the statistical validity of this test is questionable. Instead of this test, other test, such as Tukey-Kramer multiple range test, should be applied. Some results may change due to changes in statistical analysis.

3. Values for ethylene production, ACS, and ACO activity are 1000 times higher than typical values. It is necessary to check whether the values shown in the results are correct.

Overall, this manuscript has valuable findings. In my conclusion, this manuscript is suitable for publication in horticulturae if the above issues are adequately addressed.

Author Response

In this manuscript, the authors showed that BABA delayed the senescence of strawberry fruits by regulating metabolism of NO, H2S, ascorbic acid, and ABA. These results contribute to our understanding postharvest physiology of strawberry fruits. However, there are some issues in this manuscript as described below.

  1. In Introduction, it is necessary to describe the current knowledge regarding the involvement of ethylene and ABA in the aging of strawberry fruits.

       Response: So appreciate your professional suggestion. We have supplemented detailed information of ethylene and ABA in the aging of strawberry fruits. Such modification made the statements of present study more readability.

  1. Duncanʼs multiple range test is applied for statistical analysis. However, the statistical validity of this test is questionable. Instead of this test, other test, such as Tukey-Kramer multiple range test, should be applied. Some results may change due to changes in statistical analysis.

       Response: Thanks for your professional advice. We accepted your suggestions and conducted a re-analysis of the statistical data. We have accordingly made modifications in the revised manuscript. Such modification made the statements of present study more accurate.

  1. Values for ethylene production, ACS, and ACO activity are 1000 times higher than typical values. It is necessary to check whether the values shown in the results are correct.

       Response: Thanks you for your professional suggestions. The present issue is due to the units in the results. We accepted your suggestions and made modifications in the revised manuscript.

Overall, this manuscript has valuable findings. In my conclusion, this manuscript is suitable for publication in horticulturae if the above issues are adequately addressed.

Response: Thanks for your support. We have carefully answered your questions and revised the article according to your comments.

Besides, a few type errors and font have been corrected in revised version.

Round 2

Reviewer 1 Report

Comments and Suggestions for Authors

The authors covered most of the comments made, however some responses were partially evident in the document.

It is recommended that the photograph of the strawberries include the day of evaluation.  

Reviewer 2 Report

Comments and Suggestions for Authors

The revised manuscript has been sufficiently improved. I consider that this manuscript is suitable for publication in horticulturae.